# Emergence of behaviour in a self-organized living matter network

**Philipp Fleig[1,2], Mirna Kramar[2], Michael Wilczek[2], Karen Alim[2,3]***

[1]Department of Physics & Astronomy, University of Pennsylvania, Philadelphia, United States; [2]Max Planck Institute for Dynamics and Self-Organization, Göttingen, Germany; [3]Physik-Department and Center for Protein Assemblies, Technische Universität München, Garching, Germany

**Abstract** What is the origin of behaviour? Although typically associated with a nervous system, simple organisms also show complex behaviours. Among them, the slime mold *Physarum polycephalum*, a giant single cell, is ideally suited to study emergence of behaviour. Here, we show how locomotion and morphological adaptation behaviour emerge from self-organized patterns of rhythmic contractions of the actomyosin lining of the tubes making up the network-shaped organism. We quantify the spatio-temporal contraction dynamics by decomposing experimentally recorded contraction patterns into spatial contraction modes. Notably, we find a continuous spectrum of modes, as opposed to a few dominant modes. Our data suggests that the continuous spectrum of modes allows for dynamic transitions between a plethora of specific behaviours with transitions marked by highly irregular contraction states. By mapping specific behaviours to states of active contractions, we provide the basis to understand behaviour's complexity as a function of biomechanical dynamics.

## Editor's evaluation

We have judged that the response to the referee's residual comments are sufficient to allow this paper to proceed to publication. In particular, the detailed analysis of the mode spectrum and its relationship to behavior is novel and possibly of general use in this field. Also, the experimental data per se should be interesting to a wide spectrum of readers.

*For correspondence:
k.alim@tum.de

**Competing interest:** The authors declare that no competing interests exist.

## Introduction

Survival in changing environments requires from organisms the ability to transition between diverse behaviours (*Angilletta and Sears, 2011*; *Wong and Candolin, 2014*). In higher organisms, a plethora of neural dynamics enable this capacity, ranging from almost random to strongly correlated firing patterns of neurons (*Mochizuki et al., 2016*). Decoding the origin of behaviour from neuronal activity has been called the 'holy grail of neuroscience' (*Bando et al., 2019*), a task especially challenging given the vastly complex networks of neurons (*Berman, 2018*). Significant progress has been made by simultaneous tracking of neuronal activity and behaviour – defined as trajectories through spaces of postural dynamics – in the fruit fly *Drosophila melanogaster* (*Honegger et al., 2020*) and the nematode *Caenorhabditis elegans* (*Nguyen et al., 2016*). Behaviours of these systems have been identified as low-dimensional (*Stephens et al., 2008*) and hierarchical (*Berman et al., 2016*).

While these discoveries have advanced our understanding of the origin of behaviour, the complexity and size of biological neural networks make the acquisition and interpretation of experimental data especially challenging. Curiously, organisms without a nervous system may offer an ideal intermediate step towards understanding behaviour. Certain non-neural organisms readily transition between a

multitude of behaviors similar in dynamic variability to that of organisms with a nervous system (***Berg and Brown, 1972***; ***Otto and Kessin, 2001***; ***McMains et al., 2008***; ***Ben-Jacob et al., 1994***; ***Ben-Jacob et al., 2000***; ***Wan and Goldstein, 2014***; ***Wan, 2018***) and thus provide the opportunity to study the link between the underlying biophysical process and behaviour.

A non-neural organism with an exceptionally versatile behavioural repertoire is the slime mould *Physarum polycephalum* - a unicellular, network-shaped organism (***Sauer, 1982***) of macroscopic dimensions, typically ranging from a millimeter to tens of centimeters. *P. polycephalum*'s complex behaviour is most impressively demonstrated by its ability to solve spatial optimisation and decision-making problems (***Nakagaki et al., 2000***; ***Tero et al., 2010***; ***Nakagaki and Guy, 2007***; ***Dussutour et al., 2010***; ***Reid et al., 2016***), exhibit habituation to temporal stimuli (***Boisseau et al., 2016***), and use exploration versus exploitation strategy (***Aono et al., 2014***). Recently, *P. polycephalum* was found capable of encoding memory about food source locations in the hierarchy of its body plan (***Kramar and Alim, 2021***) in a process much reminding of synaptic facilitation– the brain's way of creating memories (***Jackman and Regehr, 2017***). The generation of such rich behaviour requires a mechanism allowing not only for long-range spatial coordination but also the flexibility to enable switching between different specific behavioural states.

The behaviour generating mechanism in *P. polycephalum* are the active, rhythmic, cross-sectional contractions of the actomyosin cortex lining the tube walls (***Yoshimoto and Kamiya, 1984***; ***Ueda et al., 1986***; ***Kamiya et al., 1988***). The contractions drive cytoplasmic flows throughout the organism's network (***Iima and Nakagaki, 2012***; ***Alim et al., 2013***), transporting nutrients and signalling molecules (***Alim et al., 2017***). Cytoplasmic flow is responsible for mass transport across the organism and thereby contractions directly control locomotion behaviour (***Rieu et al., 2015***; ***Lewis et al., 2015***; ***Zhang et al., 2017***; ***Bäuerle et al., 2020***; ***Rodiek et al., 2015***).

So far, only one type of network-spanning peristaltic contraction pattern has been described experimentally (***Alim et al., 2013***; ***Oettmeier et al., 2017***). However, for small *P. polycephalum* plasmodial fragments various other short-range contraction patterns have been observed (***Lewis et al., 2015***; ***Zhang et al., 2017***) and predicted by theory of active contractions (***Bois et al., 2011***; ***Radszuweit et al., 2013***; ***Radszuweit et al., 2014***; ***Julien and Alim, 2018***; ***Kulawiak et al., 2019***). Similarly, up to now unknown complex, large-scale contraction patterns might play a role in generating the behaviour of large *P. polycephalum* networks. Furthermore, transitions between such large-scale patterns are needed to allow for switching between specific behaviours, for example taking sharp turns during migration in the absence of stimuli (***Rodiek and Hauser, 2015***).

Here, we decompose experimentally recorded contractions of a large *P. polycephalum* network of stable morphology into a set of physically interpretable contraction modes using Principal Component Analysis. We find a continuous spectrum of modes and high variability in the activation of modes along this spectrum. By perturbing the network with an attractive stimulus, we show that the resulting locomotion response is coupled to a selective activation of regular contraction patterns. Guided by these observations, we design an experiment on a *P. polycephalum* specimen reduced in morphological complexity to a single tube. This allows us to quantify the causal relation between locomotion behaviour, cytoplasmic flow rate and varying types of contraction patterns, thus revealing the central role of dynamical variability in generating different behaviours.

## Results

### Continuous spectrum of contraction modes reveals large variability in organism's contraction dynamics

To characterize the contraction dynamics of a *P. polycephalum* network, we record contractions using bright-field microscopy (***Video 1***) and decompose this data into a set of modes using Principal Component Analysis (PCA). At first, networks in bright-field images are skeletonized, with every single skeleton pixel representing the local tube intensity as a measure of the local contraction state (***Bäuerle et al., 2017***). Thus, any network state at a time $t_i$ is represented by a list of pixels, $\vec{I}^{t_i}$, along the skeleton, see ***Figure 1A*** and 'Data processing' (Appendix 1). Performing PCA on this data results in a linear decomposition of the intensity vectors $\vec{I}^{t_i}$ into a basis of modes $\vec{\phi}_i$:

$$\vec{I}^{t_i} = \sum_\mu a_\mu^{t_i} \vec{\phi}_\mu . \tag{1}$$

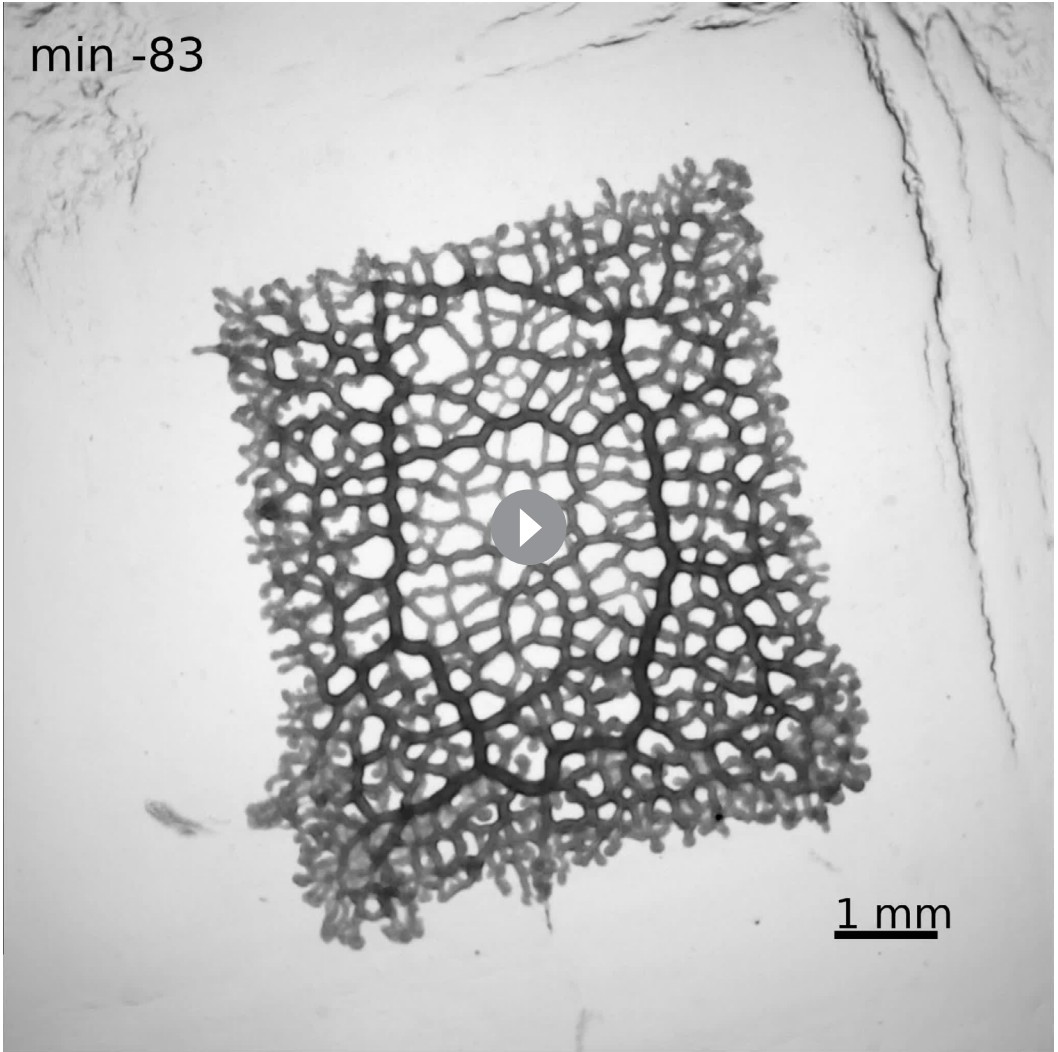

**Video 1.** Raw bright field time series of a *P. polycephalum* network, recorded at a rate of one frame every 3 sec.
https://elifesciences.org/articles/62863/figures#video1

See 'Principal Component Analysis (PCA) (Appendix 2)' for details. The modes, $\vec{\phi}_\mu$, are orthonormal eigenvectors of the covariance matrix of the data and represent linearly uncorrelated contraction patterns of the network, and $a_\mu^{t_i}$ denotes the time-dependent coefficients of the modes.

We rank modes according to the magnitude of their eigenvalues. Contrary to the small number of large eigenvalues found in a number of biological systems (*Stephens et al., 2008*; *Jordan et al., 2013*; *Gilpin et al., 2016*), here the spectrum of relative eigenvalues, see 'Principal component analysis (PCA)' (Appendix 2) for technical details, is continuous with no clear cutoff (*Figure 1B*) and as a result the contraction dynamics is high-dimensional. Notably, this is even the case when we disregard eigenvalues which lie below the upper noise bound (black line), computed from randomised data. Therefore, PCA does not directly lead to a dimensionality reduction of the data. Instead, we here investigate the characteristics of mode dynamics that result from a continuous spectrum and how these shape the organism's behaviour.

The highest-ranking modes shown in *Figure 1C(i)* have a smooth spatial structure that varies on the scale of network size. As we will discuss below, such large-scale modes are associated with the long wavelength peristalsis observed in *Iima and Nakagaki, 2012*; *Alim et al., 2013*. Interestingly, we also find modes highlighting specific morphological characteristics of the network. For example, the structure of mode $\vec{\phi}_4$, *Figure 1C(i)*, corresponds to the thickest tubes of the network *Figure 1A*, which suggests a special role of these tubes in the functioning of the network. Finally, as we go to lower

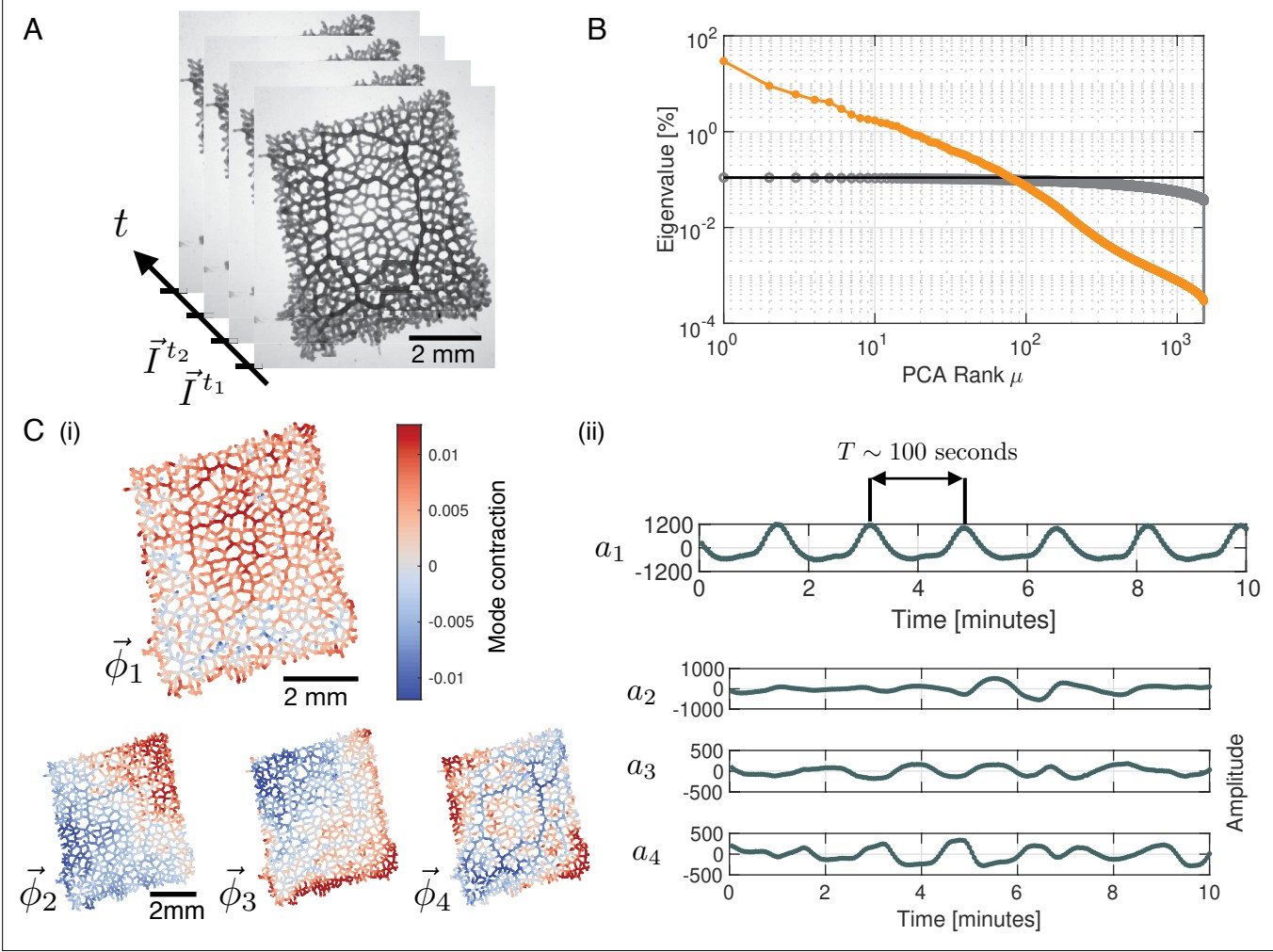

**Figure 1.** Principal Component Analysis yields a continuous spectrum of contraction modes in the *P. polycephalum* network. (**A**) Example stack of bright-field images of the recorded network. Pixel intensities encode the contraction state (tube dilation) at each point of the network. Principal Component Analysis is performed on a stack of post-processed bright-field frames. (**B**) Ranked spectrum of relative eigenvalues in percent (orange), plotted against the mode rank $\mu$ on a log-log graph. The eigenvalue spectrum is continuous, without a natural cutoff. Spectrum of randomised data (gray) shown for comparison. The cutoff for the continuous spectrum is defined by the largest eigenvalue of the spectrum from randomised data (black line). (**C**) (**i**) Structure of the four highest-ranking modes $\vec{\phi}_{1,2,3,4}$ with their respective coefficients shown in (ii). The red-blue colour spectrum indicates the contraction state. The modes are eigenvectors of the covariance matrix. The coefficient $a_1$ of the first mode captures the organism's characteristic oscillation period of $\approx 100$ sec, while the coefficients $a_{2,3,4}$ show considerable variation in amplitude and frequency over time. The PCA was performed on a data segment with 1500 frames, at the rate of 3 sec per frame.

The online version of this article includes the following figure supplement(s) for figure 1:

**Figure supplement 1.** Eigenvalue spectrum computed from randomised data.

**Figure supplement 2.** Mode amplitude dynamics over time.

**Figure supplement 3.** Spatial structure of mode $\vec{\phi}_{30}$.

ranked modes, the spatial structure of the modes becomes increasingly finer. Yet, despite lacking an obvious interpretation for their structures, like for mode $\vec{\phi}_{30}$, *Figure 1—figure supplement 1*, it is not possible to ignore their contribution relative to high-ranking modes.

Next, we turn to the time-dependent coefficients of modes shown in *Figure 1C(ii)*. In accordance with the known rhythmic contractions (*Kamiya, 1960*) the coefficient $a_1$ of the highest ranked mode $\vec{\phi}_1$ oscillates with a typical period of $T \sim 100$ sec . Most strikingly, amplitudes of mode coefficients vary significantly over time - even on orders of magnitude, as shown in *Figure 1—figure supplement 2*.

To map out the complexity of contractions over time, we define a set of *significant modes* for every time point. We quantify the activity of a mode by its *relative amplitude*

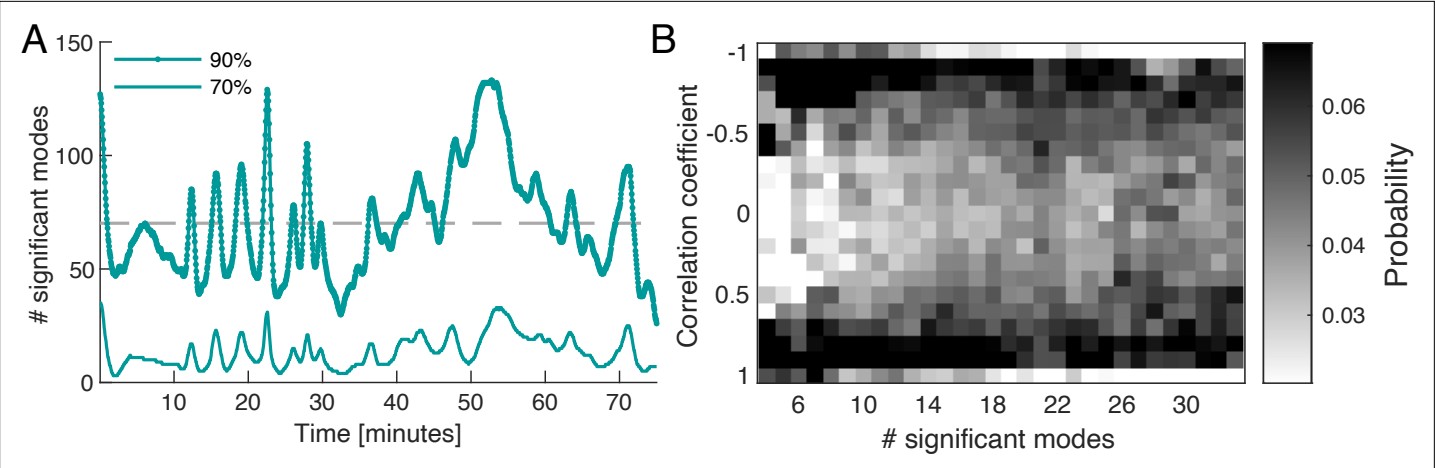

**Figure 2.** Dynamics of network contraction pattern is subject to strong variability in the percentage of significant modes and correlations between them. (**A**) Significant modes given the number of modes required for the cumulative sum of their relative amplitudes to reach 70% (thin light green) and 90% (thick dark green) of the total amplitude plotted over time. Gray dashed line is the mean value of significant modes ($\approx$ 70 modes or equivalently 4.68% of the total 1500 modes). (**B**) Distribution of temporal correlation values between mode coefficients depending on the number of significant modes taken from the 70%-cutoff curve in (**A**). Correlation values show a trend from strong (anti-)correlation for a small number of significant modes (left) to a more uniform distribution of correlation values for a large number of significant modes (right).

$$p_\mu^{t_i} = \frac{\widetilde{a^2}_\mu^{t_i}}{\sum_\nu \widetilde{a^2}_\nu^{t_i}} , \qquad (2)$$

where $\widetilde{a_\mu^2}$ denotes the amplitude of the square of the mode's coefficient. By definition the sum over the relative amplitudes of all modes is normalized to one at any given time, $\sum_\mu p_\mu^{t_i} = 1$. For any given time point, we order the modes by their relative amplitude from largest to smallest and take the cumulative sum of their values until a chosen cutoff percentage is reached, see *Figure 2A*. We find that the percentage of modes required to reach a specified cutoff value varies considerably over time. For a 90% cumulative amplitude cutoff, we find that on average 6.06% ($\approx$ 70 modes) of the 1500 modes are significant. As discussed in more detail in 'Choice of the cutoff of mode coefficient amplitudes' (Appendix 6), defining a cutoff for the cumulative sum of mode amplitudes is related to the problem of defining a cutoff for a continuous spectrum of eigenvalues. One common method is to define the cutoff with respect to the largest eigenvalue of the spectrum computed from a randomised version of the original data (*Berman et al., 2014*). In 'Choice of the cutoff of mode coefficient amplitudes' (Appendix 6), we find that the 90% cumulative amplitude cutoff considered above is consistent with this definition of cutoff for eigenvalues. As an important feature, we observe that there is large variation in the number of significant modes over time, with a standard deviation of 36.96% from the mean value. This is an indicator for the complexity of the contractions in the network.

Apart from the number of significant modes, the dynamics of the network depend on the temporal correlation of modes. While the modes form a spatially uncorrelated basis, the temporal correlation of mode activation is non-trivial. In *Figure 2B*, we show the distribution of temporal correlations between mode coefficients as a function of the number of significant modes, see 'Distribution of temporal correlations' (Appendix 3) for technical details. For a small number of significant modes, the coefficients are strongly (anti-)correlated in time, while for a large number of significant modes, correlations values between coefficients are more uniformly distributed. Here, correlated coefficients result in coordinated pumping behaviour/contractions, while least correlated coefficients coincide with irregular network-wide contractions. The above analysis shows that the dynamics of network contractions covers a wide range in complexity, from superposition of few large-scale modes strongly correlated in time, to superpositions of many modes of varying spatial scale and temporal correlations. This gives rise to strong variability in the regularity of the contraction dynamics over time. Up to now, we investigated an 'idle' network not performing a specific task, so we next stimulate the network to provoke a specific behaviour and scrutinize how the continuous spectrum of modes contributes to it.

## Stimulus response behaviour is paired with activation of regular, large-scale contraction patterns interspersed by many-mode states

To probe the connection between a specific behaviour and network contraction dynamics, we next apply a food stimulus to the same network, see *Figure 3A*. Food acts as an attractant and causes locomotion of the organism toward the stimulus in the long term. The stimulus immediately triggers the tubes in the network to grow in a concentric region around the stimulus site. Also, the thick transport tubes oriented toward stimulus location increase their volume, see *Figure 3A*. Altogether these morphological changes are typical for the specific behaviour induced here, namely the generation of a new locomotion front.

In *Figure 3B*, we quantify this stimulus response behaviour by tracking the growth of the most active regions of the network, defined by the boxes shown in the 81 min in *Figure 3A*. The tracked regions are located on opposing sides of the network. Starting approximately at 85 min, the part of the network next to the stimulus site grows rapidly (burgundy curve in *Figure 3B*), at the expense of the fan-shaped locomotion front in the lower left corner of the network (green curve in *Figure 3B*). In *Figure 3—figure supplement 2*, we additionally show that prior to the stimulus, the network grows the fan-like shaped locomotion front in the lower left corner. Taken together, the application of the stimulus leads to a reversal of the network's growth direction.

To identify potential changes in the contraction dynamics due to stimulus application, we perform PCA on a 700 frames long subset of the data subsequent to the 'idle' data of the previous section. First, we rediscover a continuous spectrum of modes, see *Figure 3—figure supplement 1*, resembling that of the 'idle' dynamic state. However, now the highest-ranked contraction modes, see *Figure 3C*, show spatial patterns which can be directly related to the network's growth behaviour. This includes activation of the upper region of the network close to the stimulus, as well as activation of the thick tubes extending from top to bottom of the network. In fact, for more than 500 frames after the stimulus has been applied, the rhythmic contraction dynamics of the network are dominated by the three highest-ranked modes, see *Figure 3D* and *Figure 3—figure supplement 3* for the oscillatory dynamics of mode coefficients. During this period, every time a single mode is the most active one for a duration of $> 30$ frames, its amplitude exceeds that of any other mode by 20–30%.

Next we link the stimulus-induced reversal in growth direction to the changes in the contraction pattern. Specifically, we observe that the time interval of growth reversal *Figure 3B* coincides with the activation of the third-ranked mode $\vec{\phi}_3$, (orange curve in *Figure 3D*), as indicated by the pink shaded box extending across *Figure 3D and B*. The structure of this mode shows a clear distinction of the growth area close to the stimulus and an activation of the two thick tubes stretching from bottom to the top of the network. This mode is followed by an activation of mode $\vec{\phi}_2$ (blue curve), clearly marking the growth region within its spatial structure.

Finally, over time the growth of the stimulus response region tapers off and we find reactivation of mode $\vec{\phi}_1$ (red curve) which was the dominant mode before stimulus application. We note that the spatial structure of mode $\vec{\phi}_1$ is remarkably similar to mode $\vec{\phi}_1$, the top-ranked mode that we find for PCA on the pre-stimulus 'idle' data *Figure 3—figure supplement 2B*. The reactivation of this mode indicates that this contraction pattern is intrinsic to the network and is not simply erased by the stimulus.

Strikingly, the regular contraction dynamics shown in *Figure 3D* are interspersed with many-mode states where the number of significant modes increases considerably, see *Figure 3E*. The number of significant modes oscillates after the stimulus. The oscillation maxima coincide with times at which the organism switches from one dominant contraction pattern to another, as indicated by the blue-shaded boxes extending across *Figure 3D and E*. Our results suggest that prolonged regular dynamics dominated by a few or even a single mode are associated with specific behaviour like locomotion and growth, while the many-mode states seem to serve as transition states between them.

While the network morphology is characteristic for *P. polycephalum*, reducing network complexity may help to conclude on the role of regular dynamics in driving specific behaviours, and the role of many-mode states and the therefrom arising continuous distribution of modes.

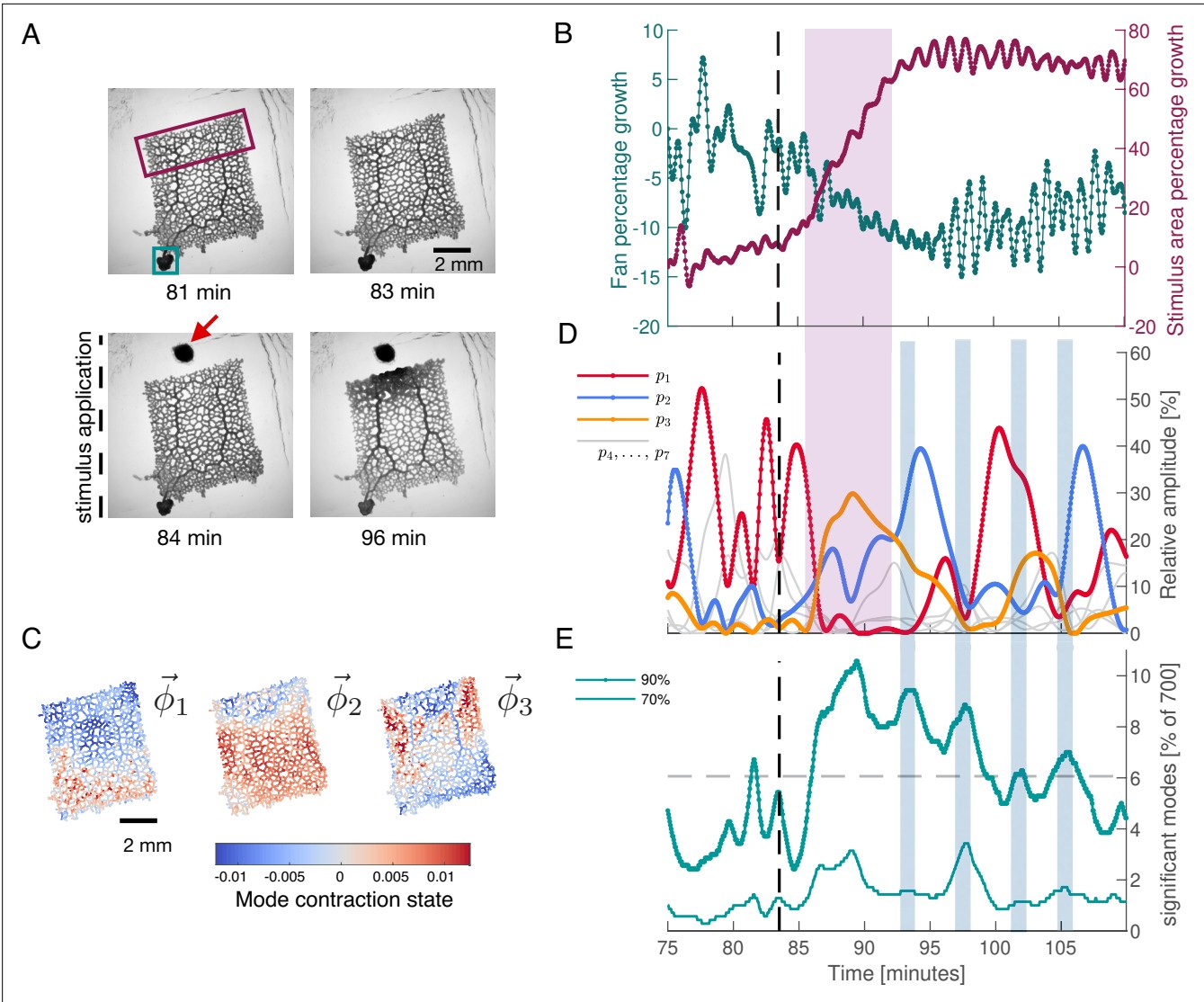

**Figure 3.** Network growth response to an external attractive stimulus is linked to characteristic changes in the contraction dynamics. (**A**) Sequence of bright-field frames showing the network's growth response to a food stimulus (red arrow in the 84 min). (**B**) Growth curves of the two most active growth regions of the network. The two tracked regions are indicated by the green and burgundy boxes in the frame at 81 min shown in (**A**). The growth is shown as the percentage change in area with respect to the initial state at 75 min. After stimulus application, the upper part of the network undergoes significant growth at the expense of the fan-like shaped locomotion front in the lower left corner. (**C**) The spatial contraction pattern of the three top-ranked modes $\vec{\phi}_1$, $\vec{\phi}_2$, and $\vec{\phi}_3$. (**D**) Activity of the three top-ranked modes measured by their respective relative amplitude, $p_\mu$. After the stimulus (dashed line at 83.5 min), time intervals with a single contraction mode dominating in amplitude (red for the relative amplitude of mode $\vec{\phi}_1$, blue for $\vec{\phi}_2$ and yellow for $\vec{\phi}_3$) prevail over all other modes. Mode amplitudes four to seven are shown in gray for reference. This growth response is paired with activation of mode $\vec{\phi}_3$, as indicated by the pink shaded box extending across (**B**) and (**D**). (**E**) Significant number of modes for a cumulatively summed amplitude of 70% (thin light green) and 90% (thick dark green), over time. Gray dashed line indicates the 6.06% ($\approx$ 42 modes) average of significant modes for the 90% criterion. When contractions switch from one dominant mode to another, we find time intervals where a larger number of modes have a similar relative amplitude. These times are indicated by the blue shaded boxes extending across (**D**) and (**E**).

The online version of this article includes the following figure supplement(s) for figure 3:

**Figure supplement 1.** Continuous eigenvalue spectrum compared to randomised spectrum.

**Figure supplement 2.** Pre-stimulus link between growth behaviour and contraction patterns.

**Figure supplement 3.** Temporal dynamics of the coefficients of the three top-ranked contraction modes.

**Figure supplement 4.** Temporal dynamics of instantaneous mode rank in a pre-stimulus part of the data.

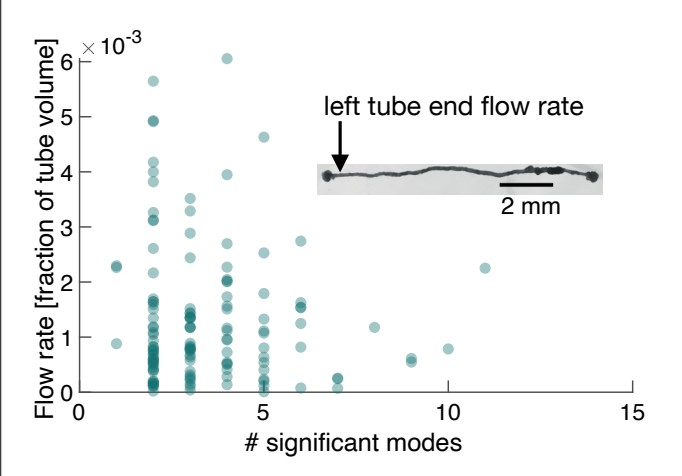

**Figure 4.** Number of significant modes is indicative for the volume flow rate in a cell reduced in its network complexity to a single tube. Inset: Single tube with locomotion fronts at both ends. Main plot: Volume flow rate at the left tube end, calculated from tube contraction dynamics versus the number of significant modes at different times. High flow rates are only achieved for a small number of significant modes.

The online version of this article includes the following figure supplement(s) for figure 4:

**Figure supplement 1.** Flow rate at the right end of the tube.

## Number of significant modes determines maximum cytoplasmic flow rate in the minimal morphological representation of the network

We next perform exactly the same course of experiments as before but on a *P. polycephalum* specimen reduced in complexity to a single tube with a locomotion front at either end, see inset in *Figure 4* and *Video 2*. Strikingly, when performing PCA on this specimen of simple morphology we again find a continuous spectrum of modes (*Figure 5—figure supplement 1*) and large variability, including spikes of many-mode states, in the number of significant modes (*Figure 5A*). This observation finally underlines that the continuous spectrum of modes and its variability in activation is intrinsic to the organism's behaviour, ruling out that the complexity of contraction modes only mirrors morphological complexity. Foremost, this minimal constituent of a network allows us now to directly map the effect of variations in the contraction dynamics onto behaviour.

From the experimentally quantified tube contractions, we calculate the maximal flow rate at any point along the tube (*Li and Brasseur, 1993*) and over time correlate the strength of the flow rates, driving locomotion behaviour at the tube ends, with the number of significant modes, see 'Flow rate calculation in a *P. polycephalum* cell with single-tube morphology' (Appendix 4). For both the flow rate at the left and right end of the tube, shown in *Figure 4*, and *Figure 4—figure supplement 1*, respectively, we find that large flow rates are only achieved when the number of significant modes is small. We had previously found that few significant modes are highly (anti-)correlated, whereas states with many significant modes are not, see *Figure 2B*. This observation now confirms our physical intuition that the irregularity of states consisting of many modes goes hand in hand with reduced pumping efficiency and thus unspecific behaviour. Since a small number of significant modes not necessarily always implies a large flow rate, we next turn to analyze their exact spatial structure and instantaneous temporal correlation to determine how cytoplasmic flow rates impact behaviour.

## Instantaneous coupling and selective activation of modes determine locomotion behaviour

We now demonstrate the impact of changes in the dynamics of a small number of modes on the organism's behaviour. For this, we quantify the locomotion behaviour of the single tube by tracking the area of the locomotion fronts protruding from each end of the tube over time, see *Figure 5A*. The growth curves of the tube ends are shown in *Figure 5B*. While initially the right end is protruding

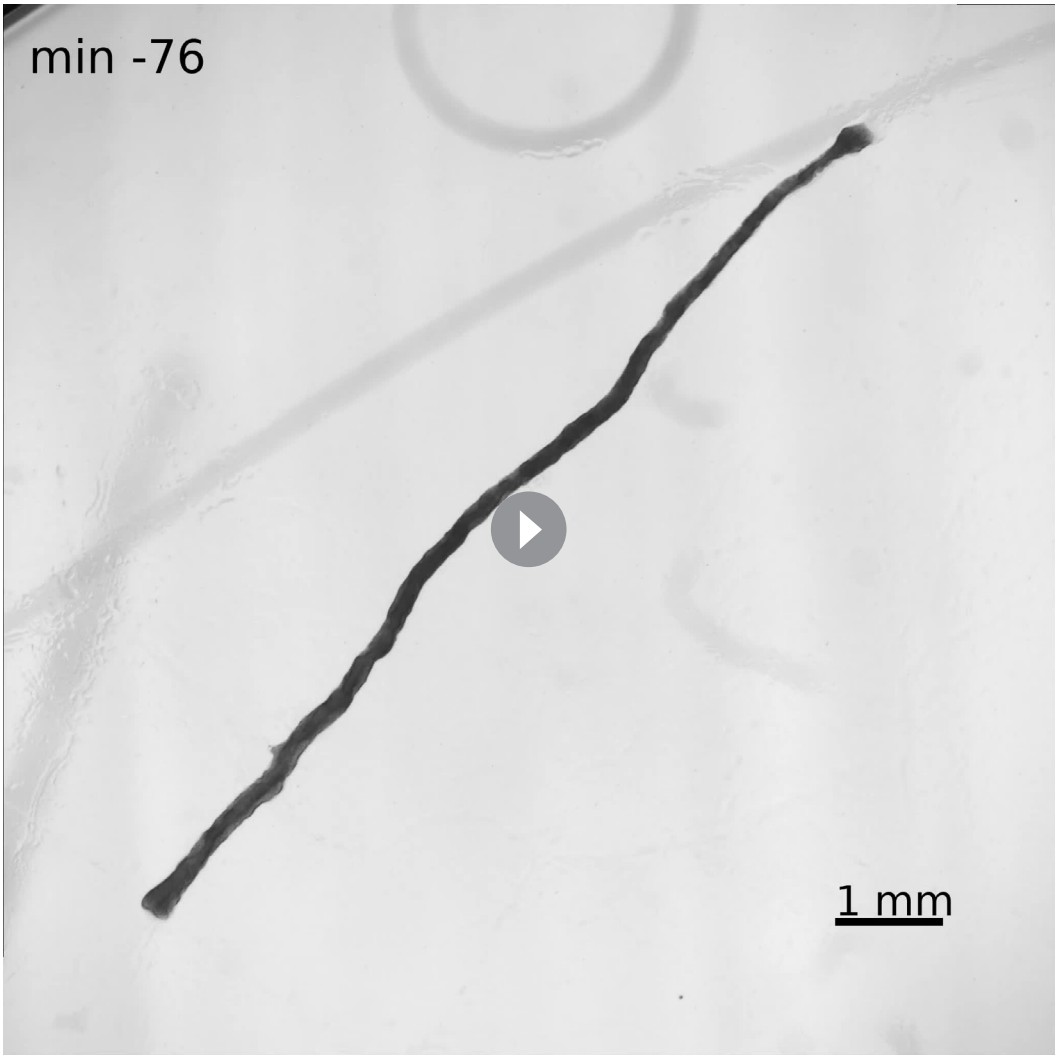

**Video 2.** Raw bright field time series of a single *P. polycephalum* tube, recorded at a rate of one frame every 3 sec.
https://elifesciences.org/articles/62863/figures#video2

faster at the expense of the left end, a food stimulus applied to the left end of the tube reverses the direction of locomotion.

As for the network, we use PCA to analyse the contraction dynamics of the single tube and link it to behaviour. We apply PCA to contraction data along the tube which we parameterize by a longitudinal coordinate. The spatial shapes of the two top-ranked modes $\vec{\phi}_1$ and $\vec{\phi}_2$ approximate Fourier modes, see *Figure 5C* and *Figure 5—figure supplement 2*. Examining the activation of modes, we find that over long time intervals, and in particular after the stimulus, the two top-ranked modes dominate the tube's contraction dynamics, see *Figure 5D*. To illustrate the connection between the nature of tube contraction dynamics and locomotion behaviour, we pick two representative time intervals after the stimulus where either only mode $\vec{\phi}_1$, or modes $\vec{\phi}_1$ and $\vec{\phi}_2$ equally, dominate overall, see vertical pink bars in *Figure 5D*. During the first interval when mode $\vec{\phi}_1$ alone is dominating, the tube is driven by a standing wave contraction pattern - yielding only a low cytoplasmic flow rate. Correspondingly, the size of the locomotion front at either end shows no significant change in area during this interval. In contrast, during the interval when both modes $\vec{\phi}_1$ and $\vec{\phi}_2$ are equally active, the resulting superposition is a left-traveling wave producing a large cytoplasmic flow rate in that direction. The left-traveling wave is in accordance with the growth of the left and retraction of the right locomotion front as quantified in *Figure 5B*. See 'Mode superpositions in a *P. polycephalum* cell with single-tube morphology'

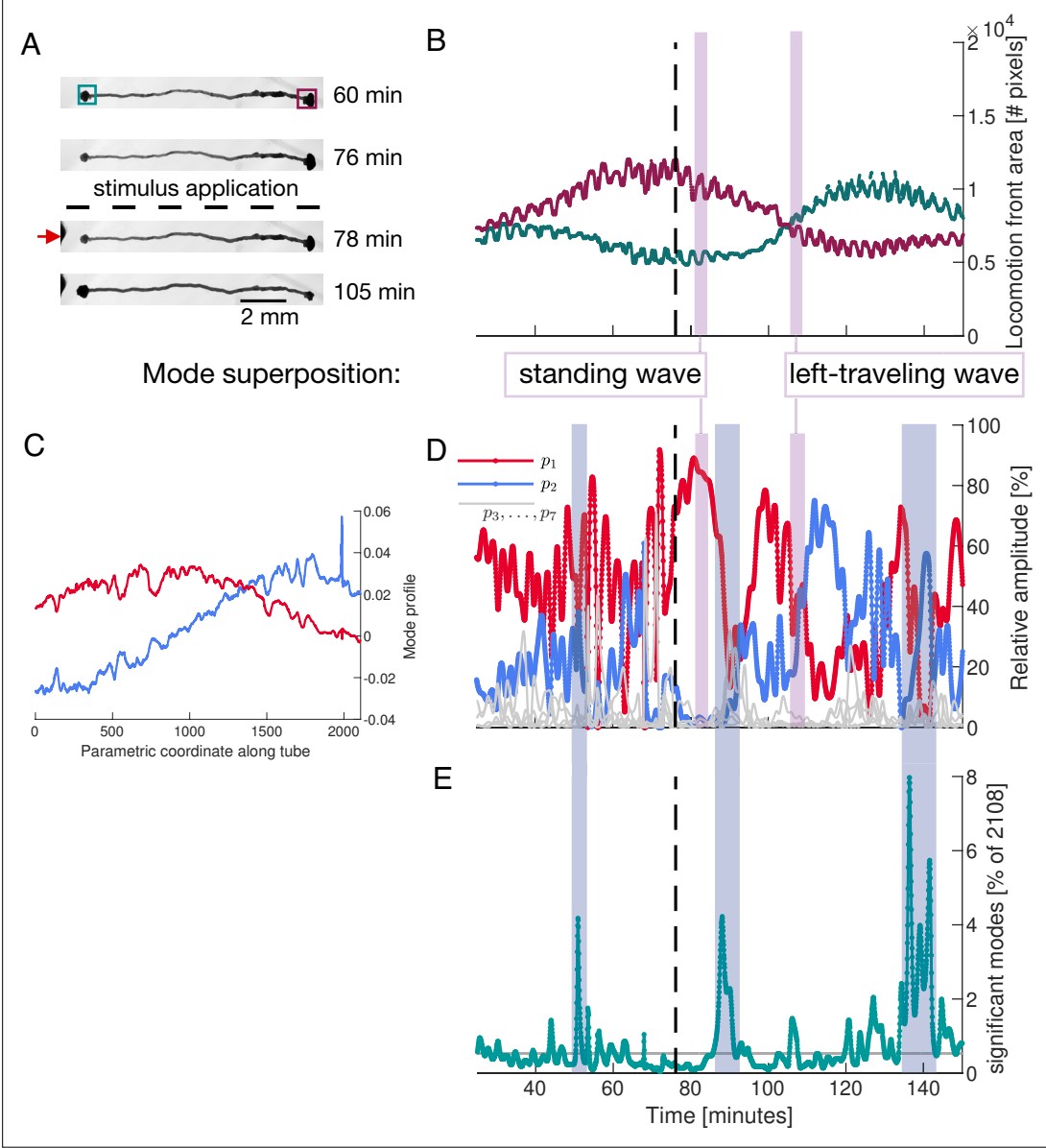

**Figure 5.** Locomotion behaviour of a single tube is determined by activation and temporal coupling of sine-and cosine-shaped contraction modes. (**A**) Sequence of bright-field images showing the locomotion behaviour of the single tube including its response to stimulus application at the left end (red arrow) at 77 min (dashed line). (**B**) Behaviour of the locomotion front at each end of the tube over time. Tracked regions of the tube are indicated by the green and burgundy boxes in top bright-field frame in (**A**). (**C**) Spatial profile of the top-ranked modes $\vec{\phi}_1$ and $\vec{\phi}_2$ approximately showing sine and cosine shape, respectively. Larger version of the plot is shown in *Figure 5—figure supplement 2*. (**D**) Activation of the two top-ranked modes given by their relative amplitude (red and blue). Relative amplitudes of lower ranked modes are shown in gray for comparison. Vertical pink boxes extending across (**B**) and (**D**) indicate two representative time intervals and the nature of the two-mode superposition is specified. (**E**) The number of significant modes over time with 90% cumulative relative amplitude cutoff. Blue boxes extending across (**D**) and (**E**) highlight the most pronounced many-mode states.

The online version of this article includes the following figure supplement(s) for figure 5:

**Figure supplement 1.** Continuous eigenvalue spectrum.

**Figure supplement 2.** Profile of contraction modes along the single tube.

**Figure supplement 3.** Temporal dynamics of the coefficients of the two top-ranked contraction modes $\bar{\phi}_1$ (red) and $\bar{\phi}_2$ (blue) in the single-tube *P. polycephalum* network.

(Appendix 4) for more details. In *Figure 5E*, we highlight the most pronounced many-mode states during changes of dominant contraction dynamics.

These two examples solve the conundrum of *Figure 4*, which shows that a small number of significant modes does not necessarily lead to high cytoplasmic flow rates. Yet, the direct mapping of contraction dynamics onto ensuing cytoplasmic flows confirms that a small number of significant modes is associated with specific behaviour. High cytoplasmic flow rates at the tube ends drive locomotion, while lower flow rates likely lead to other behaviours such as mixing. Furthermore, many-mode states seem necessary for transitions in a multi-behavioural space.

Our explanation of behaviour – from contractions via flows to locomotion behaviour – in the single tube is a template for an analogous explanation in the network morphology. The analogy is justified by the strong resemblance of the continuous mode spectrum, dynamics of significant modes, activation of regular contraction patterns and the nature of growth behaviour in both the network and single tube. Therefore, while it is beyond the scope of this study, we expect a detailed analysis of the link between contractions and flows in the network morphology to yield qualitatively similar results to those of the single tube, thus completing the mechanism of behaviour generation.

## Discussion

To uncover the origin of behaviour in *P. polycephalum,* we quantified the dynamics of this living matter network and linked it to its emerging behaviour. The simple build of this non-neural organism allows us to trace contractions of the actomyosin-lined tubes, compute cytoplasmic flows from the contractions and finally link these dynamics to the emerging mass redistribution and whole-organism locomotion behavior. Decomposing the contractions across the network into individual modes, we discover a large intrinsic variability in the number of significant modes over time along a continuous spectrum of modes. By triggering locomotion through application of a stimulus, we identify that states with few significant modes and regular contraction patterns correspond to specific behaviors, in this case locomotion. Yet, irregular contraction patterns consisting of a large number of significant modes are also present, particularly marking the transitions between different regular contraction states. The use of an organism with a single-tube morphology allows us to obtain quantitative insights into the mechanism connecting contraction dynamics and locomotion behavior and in first approximation serves as an analogue system for the large *P. polycephalum* with network morphology. Our findings suggest that a continuous spectrum of contraction modes allows the living matter network *P. polycephalum* to quickly transition between a multitude of behaviours using the superposition of multiple contraction patterns.

Networks are ubiquitous in biology, including examples such as ecological networks (*García Martín and Goldenfeld, 2006*) and biomolecular interaction networks (*Albert, 2005*). Measurable quantities of these networks, for instance the degree distribution of the network, typically follow continuous distributions and are oftentimes power-laws. The spectrum of eigenvalues *Figure 1B* that we find for the contraction dynamics in *P. polycephalum* may similarly suggest a power-law. However, the presence of a power-law is generally difficult to prove and interpret. Instead, our sole focus is on the continuous nature of the spectrum. It is important to emphasise that the continuity of the eigenvalue spectrum is not simply the result of the organism's complex network morphology. This is demonstrated by the fact that we find a similar spectrum also for the single-tube morphology *Figure 5—figure supplement 1*. Therefore, here the continuous spectrum of eigenvalues is distinctively a property of the *dynamic* state of the organism.

Our observation of interlaced regular and irregular contraction patterns in *P. polycephalum* reminds of the strongly correlated or random firing patterns of neurons in higher organisms (*Mochizuki et al., 2016*). In neural organisms, stereotyped behaviours are associated with controlled neural activity, as for example for locomotion in *C. elegans* (*Liu et al., 2018*) or the behavioural states of the fruit fly *Drosophila melanogaster* (*Berman et al., 2014*; *Berman et al., 2016*). Variability in the dynamics of behaviour is also widely observed in these neuronal organisms (*Grobstein, 1994*; *Renart and Machens, 2014*; *Werkhoven et al., 2021*; *Honegger et al., 2020*; *Ahamed et al., 2020*). It is thus likely that the transition role of irregular states consisting of many significant modes observed here for *P. polycephalum* parallels the mechanisms of generating behaviour in the more complex forms of life.

*P. polycephalum* is renowned for its ability to make informed decisions and navigate a complex environment (*Nakagaki et al., 2000*; *Tero et al., 2010*; *Nakagaki and Guy, 2007*; *Dussutour et al., 2010*;

*Reid et al., 2016*; *Boisseau et al., 2016*; *Aono et al., 2014*; *Ueda et al., 1976*; *Miyake et al., 1991*). It would be fascinating to next follow the variability of contraction dynamics during more complex decision-making processes. Furthermore, it would be interesting to observe 'idle' networks during foraging over tens of hours. It is likely that the contraction states with many significant modes here act as noisy triggers that can spontaneously cause the organism to reorient its direction of locomotion.

In the context of *P. polycephalum*'s foraging behaviour, another exciting line of research opened by our results is the link between contraction modes and the organism's metabolic changes. The foraging networks displays a plethora of morphological patterns which are linked to the underlying metabolic states (*Takamatsu et al., 2017*; *Lee et al., 2018*). It has recently been shown that in the neural organism *Drosophila melanogaster*, behaviour stemming from neural activity causes large-scale changes in metabolic activity (*Mann et al., 2021*). Exploring the relationship between behaviour emergence and metabolism in *P. polycephalum* will bring key insight about the interplay between the mechanical and the biochemical machinery of the organism.

*P. polycephalum*'s body-plan as a fluid-filled living network with emerging behaviour finds its theoretical counterpart in theories for active flow networks developed recently (*Woodhouse et al., 2016*; *Forrow et al., 2017*). Strikingly, these theories predict selective activation of thick tubes which we observe in the living network as well, prominently appearing among the top ranking modes, see $\vec{\phi}_4$ in *Figure 1C(i)* or $\vec{\phi}_3$ in *Figure 3C*. This is a first hint that dynamics states arising from first principles in active flow networks could map onto behavioural and transition states observed here.

Likely our most broadly relevant finding in this work is that irregular dynamics, here arising in states with many significant modes, play an important role in switching between behaviours. This should inspire theoretical investigations to embrace irregularities rather than focusing solely on regular dynamic states. The most powerful aspect of *P. polycephalum* as a model organism of behaviour lies in the direct link between actomyosin contractions, resulting in cytoplasmic flows and emerging behaviours. The broad understanding of the theory of active contractions (*Bois et al., 2011*; *Radszuweit et al., 2013*; *Radszuweit et al., 2014*; *Julien and Alim, 2018*; *Kulawiak et al., 2019*) might therefore well be the foundation to formulate the physics of behaviour not only in *P. polycephalum* but also in other simple organisms. This would not only open up an new perspective on life but also guide the design of bio-inspired soft robots with a behavioural repertoire comparable to higher organisms.

## Materials and methods
### Experiments
The specimen was prepared from fused microplasmodia grown in a liquid culture (*Daniel et al., 1962*) and plated on 1.5%-agar. The network was trimmed and imaged in the bright field setting in Zeiss ZEN two imaging software with a Zeiss Axio Zoom V.16 microscope equipped with a Hamamatsu ORCA-Flash 4.0 digital camera and Zeiss PlanNeoFluar 1 x/0.25 objective. The acquisition frame rate was 3 sec. The stimulus was applied in a form of a heat-killed HB101 bacterial pellet in close network proximity.

## Additional information

### Funding

| Funder | Grant reference number | Author |
| --- | --- | --- |
| Simons Foundation | 400425 | Philipp Fleig |
| IMPRS for Physics of Biological and Complex Systems | | Mirna Kramar |
| Max Planck Society | | Philipp Fleig<br>Mirna Kramar<br>Michael Wilczek<br>Karen Alim |

| Funder | Grant reference number | Author |
|---|---|---|

The funders had no role in study design, data collection and interpretation, or the decision to submit the work for publication.

## Author contributions

Philipp Fleig, Conceptualization, Data curation, Formal analysis, Investigation, Methodology, Software, Validation, Visualization, Writing – original draft, Writing – review and editing; Mirna Kramar, Data curation, Investigation, Resources, Writing – review and editing; Michael Wilczek, Methodology, Writing – review and editing; Karen Alim, Conceptualization, Funding acquisition, Investigation, Project administration, Resources, Supervision, Writing – original draft, Writing – review and editing

## Author ORCIDs

Philipp Fleig (iD) http://orcid.org/0000-0003-4103-7478
Mirna Kramar (iD) http://orcid.org/0000-0002-1637-140X
Michael Wilczek (iD) http://orcid.org/0000-0002-1423-8285
Karen Alim (iD) http://orcid.org/0000-0002-2527-5831

## Decision letter and Author response

Decision letter https://doi.org/10.7554/eLife.62863.sa1
Author response https://doi.org/10.7554/eLife.62863.sa2

# Additional files

## Supplementary files

• Transparent reporting form

## Data availability

The two datasets from which Figures 1,2 and 3 and Figures 4 and 5 were generated are included as videos of raw bright-field time series in the article.

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

# Appendix 1

## Data

The typical thickness of tubes in a *P. polycephalum* network is $\sim 50-100\mu$m and the contraction amplitude about ~10% of the tube's typical thickness (*Alim et al., 2013*). This change in tube thickness can be detected from a bright-field microscopy recording. We record one bright-field frame every three seconds. Since the periodic contractions of the tubes take place on the time scale of 100 sec, they are thus well resolved by the selected frame rate. Typically an idle network keeps a stable morphology and does not move significantly over a period of 1.5 h to 2.5 h which we use for recording its contraction dynamics. Since no two *P. polycephalum* specimens ever have the same network morphology, we are naturally constrained to one biological and one technical replicate in our experiments.

## Data processing

Our data is a stack of bright-field images recorded from the *P. polycephalum* network with a rate of one frame every three seconds (*Video 1*; *Video 2*). Each bright-field frame has a time label $t_i$ and the total number of frames is given by $T$. We process this data in the following steps. First, we mask the network in the bright-field images through thresholding. It is important to note that we use the same mask for all the images in the stack. This is possible since we consider a network that does not significantly move or change its morphology. This is true even when we apply a stimulus to the network, since we only consider the initial stages of stimulus response, before the network starts to display strong movement. From the masked regions of the bright-field frames, we extract pixel intensity values which we convert to 8-bit format. Since we are here primarily interested in the contraction dynamics of the organism and not in the the actual base thickness of tubes or its long-term growth dynamics, we detrend the data using a moving-average filter (rational transfer function) with a window size of two contraction periods (~ 200 sec) (*Bäuerle et al., 2017*). This leaves us only with the desired information about contractions taking place in the time scale of several minutes. We store the intensity values of each frame in a vector $\vec{I}^{\,t_i}$ of dimension $M$ equal to the number of pixels in the network, and  indexes the frames in the range $i = 1, \dots, T$. From the post-processed data, we define the following data matrix

$$\mathbf{X}^t = \left( \vec{I}^{\,t_1}, \vec{I}^{\,t_2}, \dots, \vec{I}^{\,t_T} \right) , \tag{3}$$

where $t$ denotes the matrix transpose.

## Appendix 2

### Principal component analysis (PCA)

The contraction modes are computed from the covariance matrix of the data. We compute the covariance matrix from the data matrix $\mathbf{X}$ after subtracting the mean from each column. The covariance matrix is given by

$$p_\mu^{t_i} = \frac{\widetilde{a^2}_\mu^{t_i}}{\sum_\nu \widetilde{a^2}_\nu^{t_i}} \,, \tag{4}$$

The sought after contraction modes $\vec{\phi}_\mu$ are the eigenvectors of the covariance matrix

$$\mathbf{C}\vec{\phi}_\mu = \lambda_\mu \vec{\phi}_\mu \,, \tag{5}$$

and $\lambda_\mu$ is the eigenvalue. The number of non-zero eigenvalues is equal to the rank of the covariance matrix. The eigenvalue captures the variance of the data along the direction of mode $\vec{\phi}_\mu$. We also define the relative eigenvalue as

$$\tilde{\lambda}_\mu = \frac{\lambda_\mu}{\sum_{\nu=1}^{T} \lambda_\nu} \,. \tag{6}$$

The mode coefficient $a_\mu$ is obtained by projecting the data onto mode $\vec{\phi}_\mu$.

We note that we perform PCA on data segments with at least 700 frames (=35 min). Since it is well known from the literature (*Kamiya, 1960*) that the period of the contraction dynamics in P. polycephalum is on the order of 100 sec the analysed data contains on the order of 20 contraction periods at minimum. We can therefore be sure that we use enough data to resolve the characteristic dynamical features investigated here. As a further reassuring result, we recover the typical contraction period of 100 sec in our analysis, see *Figure 1C(ii)*.

We add a brief comment on Fourier analysis, as alternative decomposition method to PCA. First, in one dimension, PCA is equivalent to Fourier decomposition. This is indeed apparent in our PCA analysis of the single-tube data set where the principal components shown in *Figure 5C* correspond precisely to half a period of a sine and cosine Fourier mode. In two dimensions, the situation is more complicated. While we could in principle apply 2D Fourier decomposition we would need to apply Fourier analysis separately to every frame in our data set. However, this would mean that we have no information about the temporal evolution of mode activation. The Fourier modes would be different from one frame to the next and the activation of large-scale patterns over time would be obscured.

## Appendix 3

### Distribution of temporal correlations

For a given time point $t_i$, the significant modes are determined based on the 70% criterion curve from *Figure 2A*. Next, the temporal correlations among the coefficients are computed in a time interval of ±15 frames around the time point $t_i$. The correlations are then counted in bins of the appropriate row of *Figure 2A*. Repeating this processing for all time points and normalising each row by the total number of correlations in that row, we obtain the final distribution shown.

## Appendix 4

### Flow rate calculation in a *P. polycephalum* cell with single-tube morphology

To compute the flow rate of the cytoplasm in a *P. polycephalum* specimen with single-tube morphology we use the theory developed in *Shapiro et al., 1969*; *Li and Brasseur, 1993*. In that work the flow of an incompressible Newtonian fluid inside an axisymmetric tube of fixed length is considered and the equations for the flow velocity field are written in the lubrication theory approximation. Furthermore, a time-dependent thickness profile of longitudinal waves is imposed in the tube. Assuming no-slip boundary conditions, the flow field can be fully determined at every point along the tube as a function of the time-dependent tube profile. For the case when the tube profile is a periodic train of waves, we compute the volume flow rate averaged over an oscillation period by evaluating equation (13) of *Li and Brasseur, 1993*. We express the flow rate in units of volume of the entire tube divided by the oscillation period. This serves to characterize the performance in pumping of the significantly contracting *P. polycephalum* cell. We determine the time period over which to average the volume flow rate directly from the flow-rate curve. Furthermore, the thickness profile of the tube is given by the measured pixel intensity profile.

## Appendix 5

### Mode superpositions in a *P. polycephalum* cell with single-tube morphology

We are interested in how the contraction dynamics of the cell controls the cell's locomotion behavior. In our analysis we therefore focus on the tube segment connecting the locomotion fronts at either end of the tube and perform Principal Component Analysis only on this part of the cell. Since the tube is effectively one-dimensional, we find that the modes we obtain closely approximate Fourier modes. This means that superpositions of these modes afford a clear interpretation in terms of different contraction-wave patterns. Such an interpretation is even further facilitated by the fact that we find that over large time intervals after the stimulus, the number of significant modes is very small. Indeed, over such time intervals it is sufficient to approximate the contraction dynamics with only one or two modes, as can be seen from *Figure 5D*. Hence we are essentially studying a superposition of modes $\vec{\phi}_1$ and $\vec{\phi}_2$ shown in *Figure 5C* and *Figure 5—figure supplement 2* of the main text with their oscillating mode coefficients shown in *Figure 5—figure supplement 3*. To develop intuitive understanding of the nature of the superposition, we note that the modes $\vec{\phi}_1$ and $\vec{\phi}_2$ approximate sine and cosine functions over the length of the tube. Given a sine and cosine spatial contraction profile, different types of superpositions can be formed depending on the nature of their time-dependent coefficients. To illustrate further, let us assume the idealised case where both coefficients are sine functions that can have different phases and amplitudes. Then, if the coefficient of one contraction profile is very small compared to the other, the resulting superposition is a standing wave. In case the coefficients have equal amplitudes, but are phase shifted by $\pi/2$, the superposition is a traveling waves. Finally, if the coefficient amplitudes are not equal and the phase shift lies somewhere between zero and $\pi/2$, the nature of the superposition is a mix of standing and traveling wave. Extrapolating this idealised picture allows us to infer the contraction dynamics resulting from our two-mode approximation. We see that the coefficients of the two modes $\vec{\phi}_1$ and $\vec{\phi}_2$ shown in *Figure 5—figure supplement 3* change in amplitude and phase relative to each other. It is easy to identify from this plot together with the plot of relative amplitudes in *Figure 5D* time intervals which approximate one of the contraction dynamics that we have described for the idealised system. Therefore we conclude that the superposition of the two top modes changes in its nature over time, ranging form a pure standing wave to a pure traveling wave.

## Appendix 6

### Choice of the cutoff of mode coefficient amplitudes

Our analysis of contraction dynamics requires us to place a cutoff on the amplitude of mode coefficients. Our chosen cutoff of 90% is supported in two ways:

First, the problem of choosing a cutoff for the coefficients is related to the problem of choosing a cutoff for the eigenvalue spectrum, since an eigenvalue is the variance of the mode coefficient. Given the continuous nature of the eigenvalue spectrum, there is no unique way to choose a cutoff. In (*Berman et al., 2014*), it is proposed to define the cutoff by the largest eigenvalue of the spectrum of the randomised data, see the black line in *Figure 1B*. We tested their criterion on our data and find a 93% cutoff, equivalent to roughly 70 modes. This is consistent with our choice of a 90% cutoff for the amplitudes.

Second, our main qualitative observation - considerable variation in the number of significant modes over time - is robust to different choices of cutoff values. In *Figure 2A* and similarly in *Figure 3E*, we show the number of significant modes for two different values of the cutoff, namely 70% and 90%.

