## [Editor Report]

We have judged that the response to the referee's residual comments are sufficient to allow this paper to proceed to publication. In particular, the detailed analysis of the mode spectrum and its relationship to behavior is novel and possibly of general use in this field. Also, the experimental data per se should be interesting to a wide spectrum of readers.

---

## [Decision Letter]

**Decision letter after peer review:**

Thank you for submitting your article "Emergence of behavior in a self-organized living matter network" for consideration by *eLife*. Your article has been reviewed by 3 peer reviewers, and the evaluation has been overseen by a Reviewing Editor and Naama Barkai as the Senior Editor. The reviewers have opted to remain anonymous.

The reviewers have discussed the reviews with one another and the Reviewing Editor has drafted this decision to help you prepare a revised submission.

Summary:

The paper on "The emergence of behavior …" has been seen by three referees all of whom have worked in the field of quantitative measures of organismal behavior. Based on their reports and my own reading of the paper, it is clear that the paper is not acceptable in its current form. As will be explained in detail below, there are many questions regarding exactly what is being claimed and proven regarding the modal analysis of the data. As there are no significant qualms about the experimental data per se and there is universal agreement that the questions being posed are important and the *Physarum* setting is a reasonable choice for addressing those questions, I have decided that the authors should be given a chance to address these questions. Only if these questions can be resolved by what undoubtedly would be a major revision would the manuscript be reconsidered for publication.

Essential revisions:

Without a more detailed presentation of the meaning of the various results shown for the eigenvalue spectrum and the model structure, it is very hard to assess the extent to which the observed changes are actually correlated with important behavioral responses. This is partially because of vague and unsupported statements such as (line 161) "This observation finally underlines that the continuous spectrum of modes and its variability in activation is intrinsic to the organism's behavior", and (line 215) "Our findings suggest that a continuous spectrum of contraction modes allows the living matter network P. polycephalum to quickly transition between a multitude of behaviors using the superposition of multiple contraction patterns."

Specific points that need to be addressed in this regard include:

– The discussion of the eigenvalue spectrum is severally lacking. While it is true that the authors don't find a low-dimensional system (although I am not sure why they expected to), they do find a power-law spectrum. There is a vast literature of graphs that have this property (including many from biology and ecology) and the authors should connect with this work. It is also true that purely random networks also have a continuous eigenvalue spectrum, although it is not a power law it can look like one sometimes. How well can the authors quantitatively determine that the observed data is a power law and not more representative of a random process. Is there some specific control dataset that can be generated by randomization of the original data to demonstrate the biological meaning of the features they find?

– It is difficult to judge if the findings from the tube explain the results for the networks. This is due to the fact that different aspects of the PCA are shown for different cases. Some problems are (i) is the symmetry breaking seen in the tube also present in the network and how can we see it (e. g. in the shape of the modes) and (ii) do the numbers of significant modes change in the idle tube in a similar way before signal application vary in a similar fashion than in the network.

– The authors should carefully bring out the relation between qualitative change of dynamics and the behavior (i. e. response to external stimulus) more clearly to justify title of their study. The authors seem to have shown that the behavior is realized by a change in contraction dynamics (one form of self-organisation) in an otherwise relatively constant random network topology (another form of self-organisation). Given that the self-organisation in the topology of Physarum has been studied abundantly in previous work (e. g. Nakagaki et al. in Science …), the novelty here is the claim that the dynamics correspond to different "behaviors" such as moving towards or away from a stimulus or enhanced pumping of optical fluid.

[Editors' note: further revisions were suggested prior to acceptance, as described below.]

Thank you for resubmitting your work entitled "Emergence of behavior in a self-organized living matter network" for further consideration by *eLife*. Your revised article has been evaluated by Naama Barkai (Senior Editor) and a Reviewing Editor.

All reviewers and I agree that the manuscript has been significantly improved but there are some remaining issues that need to be addressed, as outlined below:

As can be seen from the detailed reviews given below, there is still a somewhat mixed opinion about exactly how informative the mode analysis is regarding behavior. There are also some residual technical questions that could be answered in a modest revision. Given these reports and a positive recommendation from an initial third reviewer, I expect that the paper will be acceptable for publication once these questions are properly addressed.

*Reviewer #1:*

Overall, the manuscript has improved in revision and I appreciate the efforts of the authors. There are, however, still rough patches and I list some specific comments below.

I am still left wanting for more understanding of the continuous spectrum. Perhaps control is just spatially local? How dependent is the finding of high-dimensionality on the particular representation of the behavior local contractions0? Or on the method (PCA)?

Line 16

I recommend removing "Surprisingly" as it is not clear why we would actually expect a low-dimensional covariance space. I would also remove an additional occurrence of "surprisingly" in the last paragraph of the introduction.

Line 26

The choice of Marom et al. (2002) as a reference is odd as this refers to neural activity in cultured networks, that is those without a body and thus without behavior.

Line 37

I would consider adding more recent references from work on ciliated organisms, e.g. work from the KY Wan and R Goldstein groups.

Line 103 How do we know that phi_4 is really preferentially exciting thicker tubes? A single image could simply be a spatial accident. Is there a correlation over the whole dataset?

Line 129 The temporal correlation of the modes should be better described. A first glance this is very confusing as the modes from PCA are, of course, uncorrelated across the whole dataset. Thus these correlations are coming from the window of +- 15 frames. This should be clarified in the main text and/or the caption for Figure 2.

Line 151

Why do we jump from a description of Figure 3A immediately to Figure 3E and then later come back to B-D?

*Reviewer #3:*

While the revised manuscript is improved in several ways, I remain unconvinced that the author's PCA-based analysis has revealed substantially new understanding about the behavior of P. Polycephalum. My main criticisms are the following:

1. As the authors mention in the paper, there are many ways to get the observed continuous power spectrum. E.g., pure 1/f noise would suffice. As such, this seems like a negative result, and I am not sure it adds to our understanding of the system.

2. The PCA modes appear to be approximately combinations of 2D plane waves, as is expected for a general 2D system. Indeed, for most 2D constructions without specific enhanced structures, a Fourier-like decomposition can be shown to be optimal. Given the increasing wavenumber for increasing mode number (as the authors defined using the power/eigenvalues), I think the data indicates that spatial frequency would be more informative compared to PCA.

3. The authors most interesting finding appears in Figure 3. relating the amplitudes in different modes and the number of "significant" modes to the application of a stimulus. I believe the authors do show that the response *is not* a simple 2-state traveling wave (peristaltic pump), but the authors don't then follow up with a description/understanding of what the response actually *is*. Apparently, the organism engages many more modes to produce the behavior. Why does this combination require so many modes and how does it work? If the response is actually linear in shape (or step shaped, or anything with many frequencies), a Fourier decomposition would involve many modes (actually an infinite number). I struggle to see how the mode decomposition as presented has taught us how the slime mold moves or responds to stimuli.

---

## [Author Response]

Essential revisions:Without a more detailed presentation of the meaning of the various results shown for the eigenvalue spectrum and the model structure, it is very hard to assess the extent to which the observed changes are actually correlated with important behavioral responses. This is partially because of vague and unsupported statements such as (line 161) "This observation finally underlines that the continuous spectrum of modes and its variability in activation is intrinsic to the organism's behavior", and (line 215) "Our findings suggest that a continuous spectrum of contraction modes allows the living matter network P. polycephalum to quickly transition between a multitude of behaviors using the superposition of multiple contraction patterns."

We thank the reviewers for pointing out the need for a more detailed presentation of the results.

We have now significantly modified the manuscript to better highlight how the changes in contraction dynamics correlate with behavioural responses, in particular with the most robust behavioural response in *Physarum*: creation of locomotion fronts upon food encounter. We modified the highlighted sentences and provided direct references to the figures depicting the described results.

Specific points that need to be addressed in this regard include:– The discussion of the eigenvalue spectrum is severally lacking. While it is true that the authors don’t find a low-dimensional system (although I am not sure why they expected to), they do find a power-law spectrum. There is a vast literature of graphs that have this property (including many from biology and ecology) and the authors should connect with this work. It is also true that purely random networks also have a continuous eigenvalue spectrum, although it is not a power law it can look like one sometimes. How well can the authors quantitatively determine that the observed data is a power law and not more representative of a random process. Is there some specific control dataset that can be generated by randomization of the original data to demonstrate the biological meaning of the features they find?

We thank the reviewers for pointing out the need for additional theoretical discussion of the eigenvalue spectrum to supplement the already existing experimental analysis of the impact that network topology has on spectrum.

First, we would like to note that while we agree that the observed eigenvalue spectrum suggests a power law, we refrain from claiming so in the article. As already mentioned by the reviewers, a power law-like eigenvalue spectrum can have many origins and is notoriously difficult to delineate, see e.g. M. Stumpf and M. Porter, (2012) “Critical Truths About Power Laws”, Science, Vol. 335, Issue 6069, pp. 665-666. The most important observation for us is that the spectrum is continuous and has no obvious cutoff value. To further corroborate that the observed structure of the eigenvalue spectrum is the result of network specific contraction dynamics, we provide two additional analyses:

We created a control data set by randomising the original data along the temporal axis. Specifically, we independently randomize the data in each column (for each spatial point) of the data matrix and then perform PCA on this randomized data. The spectrum of the randomized data, see Figure 1—figure supplement 1, shows no remnants of a power law. The comparison between original and control data shows that the original eigenvalue spectrum truly reflects contraction dynamics that underlie processes of biological meaning: sensing nutrients, feeding, and locomotion. We now depict the mode distribution of the randomized data in the manuscript as Figure 1B.

To further exclude network topology as the source of the continuous eigenvalue spectrum, we generated simulated data of a traveling wave and imposed it onto the network. The PCA analysis reveals only two modes, with their coefficients shifted in phase by 90 degrees to superimpose into a traveling wave, see Author response image 1. Thus, spatial network complexity is not the cause of the continuous eigenvalue spectrum.

**Author response image 1. sa2fig1:** PCA of traveling wave simulated data imposed on the network from the experiment. Peristaltic wave runs horizontally from right to left across the network. The pattern is fully reconstructed with two PCA modes. (A) The temporal dynamics of the mode coefficients, given by two sine waves shifted by ninety degrees with respect to each other. (B) Spatial structure of the two modes. The spatial patterns are also shifted by a quarter wavelength with respect to each other.

– It is difficult to judge if the findings from the tube explain the results for the networks. This is due to the fact that different aspects of the PCA are shown for different cases. Some problems are (i) is the symmetry breaking seen in the tube also present in the network and how can we see it (e. g. in the shape of the modes) and (ii) do the numbers of significant modes change in the idle tube in a similar way before signal application vary in a similar fashion than in the network.

We thank the reviewers for pointing out that we did not fully demonstrate the analogy between the dynamics within a network and a single tube.

In point (i), the reviewers refer to the phenomenon of ‘symmetry breaking’ in the creation of locomotion fronts. While we agree that it is tempting to use this term we prefer to characterise the observed phenomenon as a reversal of locomotion direction. The analogy between single tube and network holds independent of semantics. Namely, prior to the stimulus, the network shows the formation (growth) of a locomotion front at the bottom left corner. After the application of the stimulus, at the opposite edge of the network, the locomotion direction is reversed. We find that reversal of locomotion direction is correlated with the activation of distinct contraction patterns, see Figure 3 (post-stimulus) and Figure 3—figure supplement 2 (pre-stimulus). Modes 2 and 3 in Figure 3 B are the drivers of locomotion front generation post-stimulus. We also note the activation of mode 1 roughly 20 minutes after the stimulus application (see Figure 3 D). Notably, mode 1 has an almost identical spatial structure to mode 1 pre-stimulus shown in Figure 3—figure supplement 2B. The reactivation of this mode indicates that this contraction pattern is intrinsic to the network and is not simply erased by the stimulus.

The contraction dynamics associated with locomotion front reversal is analogous in the tube and the network: post-stimulus, the original contraction pattern is dominated by a new mode, but with the possibility to switch back to the original pattern. The one-dimensional nature of the single tube additionally allows us to develop a mechanistic understanding of locomotion in terms of standing and traveling wave contraction patterns.

In point (ii) the reviewers are asking whether there is a similarity between the dynamics of the number of significant modes in the single tube and in the network pre-stimulus. In Author response image 2 we show the relevant curve of both systems over equal length time intervals before the stimulus. During these intervals, both the single tube and the network experience slow, directed growth. In both cases the number of significant modes shows a considerable variation over time.

**Author response image 2. sa2fig2:** Dynamics of the number of significant modes in the network (A) and single tube (B) before application of the stimulus. Dashed line is the mean of the curve. Common to both systems is the large variability in the number of significant modes.

To demonstrate the analogy in contraction dynamics between network and single tube we now include new Figure 3 with Figure 3-supplement 2, respectively, in the revised manuscript and we explain these results in section “Stimulus response behavior is paired with activation of regular, large-scale contraction patterns interspersed by many-mode states”.

– The authors should carefully bring out the relation between qualitative change of dynamics and the behavior (i. e. response to external stimulus) more clearly to justify title of their study. The authors seem to have shown that the behavior is realized by a change in contraction dynamics (one form of self-organisation) in an otherwise relatively constant random network topology (another form of self-organisation). Given that the self-organisation in the topology of Physarum has been studied abundantly in previous work (e. g. Nakagaki et al. in Science …), the novelty here is the claim that the dynamics correspond to different “behaviors” such as moving towards or away from a stimulus or enhanced pumping of optical fluid.

The reviewers are correct: we use the decomposition of tube contractions to show that behaviour is the result of a rich repertoire of contraction states. For the first time, the attention is drawn away from the topological changes in the network and instead focused on the processes that drive them. In Figure R3 and R4 we correlate contraction dynamics and measured growth behaviours, finding distinct contraction patterns for different locomotion front states, before and after stimulus application. Notably, our analysis of network contractions shows different behaviours within an almost constant network topology. Specific behaviours are associated with a small number of dominant modes (see the different growth fronts in the network) while transitions between behaviours are marked by simultaneous activation of many modes. Our theoretical analysis using the spectrum of randomised data in Figure 1—figure supplement 1, further shows that not network complexity but contraction dynamics are key to behaviour.

We thank the reviewers for recognizing and summarizing the message of the article, which we now made clearer by including a quantitative presentation demonstrating the correspondence between contraction dynamics and behaviour for the network shown in updated Figure 3 with Figure 3-supplement 2.

References:

1. Stumpf, M. and Porter, M., (2012) Critical Truths About Power Laws. Science, Vol. 335, Issue 6069, pp. 665-666.

2. Berman GJ, Choi DM, Bialek W, Shaevitz JW (2014) Mapping the stereotyped behaviour of freely moving fruit flies. *J R Soc Interface* 11(99). Doi:10.1098/rsif.2014.0672.

3. Julien, J.-D. and Alim, K. (2018) Oscillatory fluid flow drives scaling of contraction wave with system size. *PNAS* October 16, 2018 115 (42) 10612-10617.

[Editors’ note: further revisions were suggested prior to acceptance, as described below.]

Reviewer #1:Overall, the manuscript has improved in revision and I appreciate the efforts of the authors. There are, however, still rough patches and I list some specific comments below.I am still left wanting for more understanding of the continuous spectrum. Perhaps control is just spatially local? How dependent is the finding of high-dimensionality on the particular representation of the behavior local contractions0? Or on the method (PCA)?

We thank the reviewer for their thoughtful feedback. In our elaborate answer in the following we first present additional data on the dynamics of the continuous spectrum before we discuss the idea of localized control and what governs the high-dimensionality of the spectrum, before we finally justify our choice of method.

To address the continuous nature of the spectrum, it is important to note that the contraction dynamics of *P. polycephalum* is not a sharp switching between a few discrete large-scale contraction states. For the contraction dynamics to change smoothly as observed by us, a continuous spectrum is required allowing for contractions over a smooth spatial scale to be activated. In Figure 3—figure supplement 4, we show the *instantaneous rank* of the top 80 modes over time for the unstimulated network. The instantaneous ranking is based on the relative amplitudes (see Equation (2) in the article) and serves as a measure of activity for a mode. We observe that modes frequently change their instantaneous rank, moving across a wide range of ranks which is only possible in a continuous spectrum.

The concept of *localised control*, as suggested by the reviewer, appears to be a question regarding the underlying mechanism by which contractions are organised across the network. We emphasize that the spectrum we observe is the immediate result of the observed phenomenology of contractions and not of the mechanism by which contractions are coordinated. The continuous spectrum of long-range patterns that we observe may result from a number of different mechanisms, for instance flow-based transport of signaling molecules [2] or hydrodynamic coupling, yet it is beyond the scope of this work to discern between these.

To address the high-dimensional nature of the contraction dynamics we start with the reviewer’s suggestion that *local contractions* are the cause of high dimensionality. To discuss this, we show in Figure 1B a modified version of a plot that was included in our previous response to the reviewers. In this figure we compare the eigenvalue spectrum of the original data to the spectrum of randomised data. As indicated in Figure 1B, the spectrum of the randomised data (gray), defines an upper noise bound (red), separating the spectrum into large eigenvalues and small eigenvalues which include noise. Before discussing the possibility of local contractions, it is useful to reiterate the interpretation of the large eigenvalues. These eigenvalues correspond to contraction modes with a large spatial scale (see e.g. Figure 3C). The patterns of these modes reflect the network’s locomotion behaviour in response to the stimulus and targeted activation of thick tubes in the network (also see [1] for a discussion of behaviour on longer time scales). Notably, the number of large eigenvalues is substantial (~80). Thus, the contraction dynamics is high-dimensional even when one disregards the part of the spectrum that lies below the upper noise bound. We now point to the continuous spectrum of large eigenvalues in the revised manuscript. While it is conceivable that one can build a model of purely local contractions which yields a continuous eigenvalue spectrum, such a model would not lead to modes showing the large-scale patterns compatible with network morphology and behaviour as we observe them. However, a contribution from local contractions to the spectrum that we observe is not excluded. For instance, it is possible that the small eigenvalues (low-ranked modes) in Figure 1B, are associated with localised contractions, for instance a single tube in the network. The interpretation of such local contractions is however strongly dependent on the network’s specific morphology and local state. We emphasize again that the presence of local contractions would not change the fact that we consistently observe long-range patterns across the network.

Finally, we address the role of the choice of decomposition method for the finding of high dimensionality. In one dimension, PCA is equivalent to Fourier decomposition (regarding this point, see also our reply to reviewer #3’s second question further below). For the single tube we are thus effectively performing Fourier decomposition and find a continuous spectrum (see Figure5—figure supplement 1). In two dimensions there are other decomposition methods which differ from PCA by the assumptions made for the decomposition basis. Changes of basis typically do not affect the dimensionality, and PCA is in fact often used as a pre-analysis step that feeds into another decomposition method to change the basis. Therefore, amongst the standard 2D decomposition methods the finding of high dimensionality holds qualitatively. We provide a discussion on the choice of method in the revised Appendix 2.

Line 16I recommend removing “Surprisingly” as it is not clear why we would actually expect a low-dimensional covariance space. I would also remove an additional occurrence of “surprisingly” in the last paragraph of the introduction.

We have removed both occurrences of the word “surprisingly”. In the abstract (line 16), we have replaced it by “notably” to emphasise that it is central to our work.

Line 26The choice of Marom et al. (2002) as a reference is odd as this refers to neural activity in cultured networks, that is those without a body and thus without behavior.

We are grateful to the reviewer for pointing this out. We agree that this choice of reference is not an optimal one given the context of the study. In the revised version, we removed this reference and replaced it with a reference [Mochizuki et al.,2016], providing an overview of neuronal activity in organisms exhibiting behaviour.

Line 37I would consider adding more recent references from work on ciliated organisms, e.g. work from the KY Wan and R Goldstein groups.

We thank the reviewer for their suggestion. We have added [Wan and Goldstein, 2014] and [Wakefield et al., 2018] as new references.

Line 103 How do we know that phi_4 is really preferentially exciting thicker tubes? A single image could simply be a spatial accident. Is there a correlation over the whole dataset?

We thank the reviewer for their question. Visual inspection of bright-field frames indeed shows that thicker tubes are a dominant feature over long time intervals of the data set. Furthermore, principal components are ranked according to their variance (the eigenvalues) which is given by the time average of the square of the mode amplitude. A mode which is activated only over a short time interval is thus very unlikely to receive a ranking as high as mode phi_4.

Line 129 The temporal correlation of the modes should be better described. A first glance this is very confusing as the modes from PCA are, of course, uncorrelated across the whole dataset. Thus these correlations are coming from the window of +- 15 frames. This should be clarified in the main text and/or the caption for Figure 2.

We thank the reviewer for making us aware of this potentially confusing point. The answer is that the PCA modes (for example the network modes shown in Figure 1C, or the tube modes shown in Figure 5C) are *spatially* uncorrelated, however their activation in time can certainly be correlated. This is what we quantify in Figure 2 and we choose windows of +-15 frames to see how this correlation changes over time. Following the reviewer’s suggestion we have clarified this point in the text and the caption of Figure 2.

Line 151Why do we jump from a description of Figure 3A immediately to Figure 3E and then later come back to B-D?

We thank the reviewer for pointing this out and we agree with them that this is a clumsy presentation. We have also noticed the same issue in Figure 5. In the revised manuscript, we rearranged the order of subplots, adjusted the letter labels in Figure 3 and Figure 5, adapted the figure captions and ensured correct referencing across the article.

Reviewer #3:While the revised manuscript is improved in several ways, I remain unconvinced that the author's PCA-based analysis has revealed substantially new understanding about the behavior of P. Polycephalum. My main criticisms are the following:1. As the authors mention in the paper, there are many ways to get the observed continuous power spectrum. E.g., pure 1/f noise would suffice. As such, this seems like a negative result, and I am not sure it adds to our understanding of the system.

We thank the reviewer for emphasising the point that a continuous spectrum can have multiple sources, including certain types of noise. With reference to our reply to reviewer #1’s first question given above (see also Figure 1B), we believe that there is enough evidence that a substantial number of the high-ranked modes are representative of large scale contraction dynamics of the network. However, this does not exclude the possibility that part of the spectrum (in particular the lower ranked modes with smaller eigenvalues) correspond to noise and possibly some form of local contractions. In fact, the possibility of noise playing an active role in shaping the network’s behaviour to us appears like an interesting feature, worthy of a future study. Regarding this point, please also see our reply to your third point below.

2. The PCA modes appear to be approximately combinations of 2D plane waves, as is expected for a general 2D system. Indeed, for most 2D constructions without specific enhanced structures, a Fourier-like decomposition can be shown to be optimal. Given the increasing wavenumber for increasing mode number (as the authors defined using the power/eigenvalues), I think the data indicates that spatial frequency would be more informative compared to PCA.

We thank the reviewer for their comment regarding Fourier decomposition as an alternative to decomposition into principal components. While the reviewer is specifically asking for the 2D case, we would first like to emphasize that in 1D, Principal Component Analysis is equivalent to Fourier decomposition. This is indeed apparent in our PCA analysis of the single tube data set where the principal components shown in Figure 5C correspond precisely to half a period of a sine and cosine Fourier mode.

In 2D, the situation is more complicated for multiple reasons. First, applying Fourier decomposition makes sense when there is a clear periodic structure visible in the data. However, from the bright-field movies we do not see spatially periodic patterns that are long-term stable. Furthermore, there are in fact enhanced structures given by the network’s morphology, such that we cannot expect waves to propagate homogeneously across the network. However, for us the most important reason for employing PCA is the ability to resolve temporal dynamics. While we could in principle apply 2D Fourier decomposition (setting our previous reasons for not doing so aside for the moment), we would need to apply Fourier analysis separately to every frame in our data set. This would mean however, that we have no information about the temporal evolution of mode activation. The Fourier modes would be different from one frame to the next and the activation of large scale patterns over time would be obscured.

For the reasons that we listed above, Principal Component Analysis is a widely used approach to analysing 2D systems with temporal dynamics. This is even the case for systems that have a simple geometry such as *C. elegans* with clear Fourier-like modes, where however the most important aspect of the analysis is how the coupling of modes evolves over time [3]. We added an explanation of the advantage of PCA over Fourier decomposition to the revised Appendix 2.

3. The authors most interesting finding appears in Figure 3. relating the amplitudes in different modes and the number of "significant" modes to the application of a stimulus. I believe the authors do show that the response *is not* a simple 2-state traveling wave (peristaltic pump), but the authors don't then follow up with a description/understanding of what the response actually *is*. Apparently, the organism engages many more modes to produce the behavior. Why does this combination require so many modes and how does it work? If the response is actually linear in shape (or step shaped, or anything with many frequencies), a Fourier decomposition would involve many modes (actually an infinite number). I struggle to see how the mode decomposition as presented has taught us how the slime mold moves or responds to stimuli.

We thank the reviewer for their question regarding the interpretation of contractions after stimulus application.

We briefly summarise the relevant observations from Figure 3. What we present in Figure 3D is that over the course of ~25min after stimulus application, a *sequence of dominant contraction patterns* is activated. These dominant patterns are given by the three highest ranked modes. Starting at minute 85, first mode 3 (orange line), then mode 2 (blue), followed by mode 1 (red), and finally again mode 2 are activated. The structure of these modes is shown in Figure 3C. The activation of these dominant patterns is interspersed by shorter time intervals where a large number of modes are roughly equally active. Additionally, in Figure 3B we quantify the behaviour of the network and find upwards locomotion behaviour. Finally, we also note that we make the analogous observations for the single tube presented in Figure 5. Thus, just like for the single tube, we find large-scale contraction patterns also for the network after stimulus application. However, due to their 2D nature and the complex network morphology, these patterns are not just simple sine and cosine waves as explained in previous point. Nevertheless, activation of specific oscillation modes can be directly linked with behaviour. This is illustrated for the activation of mode 3 by the pink box extending across subfigures Figure 3B and D. A detailed quantification of the flow rates in the network associated with specific contraction patterns is hard in complex morphologies and well beyond the scope of our study. However, given the structure of the modes one can expect them to act like pumps generating mass redistribution.

The situation is similar for the states with many significant modes marking transitions from one dominant pattern to the next. In the single tube we are actually able to correlate the number of significant modes with the magnitude of the flow rate in the tube which governs mass redistribution, see Figure 4. A large number of significant modes is linked with a small flow rate and thus small mass redistribution. In this state, modes do not add up to give a regular large-scale contraction pattern, but their superposition yields irregular patterns with a short length scale. While a similar quantification for irregular contraction states in the network is beyond the scope of this study, we believe that one of the strong points of our work is to reason by analogy with the single tube that the many mode states do not lead to mass redistribution. Additional evidence for this is given in Figure 2, where we show that when there are many significant modes, the modes are not strongly correlated in time, again suggesting that their superposition does not produce the regular large-scale patterns required for a large flow rate.

We believe that the core message of our finding is that many modes are employed to produce a broad range of behaviours, where transition between behaviours is enabled by adjustment of mode amplitude. We hope that we have now illustrated this key finding by revising Figure 3 and Figure 5 as outlined in our reply to reviewer #1’s last comment above.

References:

1. Kramar, M. and Alim, K.. Encoding memory in tube diameter hierarchy of living flow network. Proceedings of the National Academy of Sciences. 2021; 118 (10).

2. Julien, J.-D. and Alim, K. (2018) Oscillatory fluid flow drives scaling of contraction wave with system size. *PNAS* October 16, 2018 115 (42) 10612-10617.

3. Stephens, GJ., Johnson-Kerner, B., Bialek, W., Ryu, WS. Dimensionality and Dynamics in the Behaviour of *C. elegans*. PloS Computational Biology. 2008; 4(4):1-10.